# Physical Leisure Activities in Adolescents with Disabilities: Data from National Survey of Disabled Persons

**DOI:** 10.3390/healthcare12020190

**Published:** 2024-01-12

**Authors:** Mikyung Ryu, Kawoun Seo, Youngshin Song

**Affiliations:** 1College of Nursing, Chungnam National University, Daejeon 35015, Republic of Korea; rmk240@naver.com; 2Department of Nursing, Joongbu University, Chungnam 32713, Republic of Korea; kwseo@joongbu.ac.kr

**Keywords:** activities of daily living, adolescent, disabled persons, leisure activities, quality of life

## Abstract

The impact of activities of daily living (ADLs) on the life satisfaction of adolescents with disabilities and the potential role of physical leisure activities as a moderator in this context have received attention. However, little is known about the influence of physical leisure activities on the relationship between ADLs and life satisfaction in adolescents with disabilities. This cross-sectional study aimed to investigate the moderating effect of physical leisure activities on the relationship between ADLs and life satisfaction among adolescents with disabilities. Secondary data analysis was conducted by extracting data from 5364 adolescents aged 12–18 years with disabilities from the 2020 National Survey of Disabled Persons of Korea. The potential moderating effect of physical leisure activities on the relationship between ADLs and life satisfaction was investigated using Pearson’s correlation analysis and hierarchical multiple regression. Significant positive correlations were found for living, life satisfaction, and physical leisure activities. Physical leisure activities were found to play a moderating role in the relationship between ADLs and life satisfaction. This study concluded that increasing physical leisure activities, even with dependent ADLs, promotes life satisfaction, especially if ADLs are low, highlighting the necessity of increasing leisure activities in adolescents with disabilities.

## 1. Introduction

The World Health Organization (WHO) defines disability as a condition encompassing physical or mental impairments, disabilities, or social disadvantages [1]. According to Korea’s disability judgment standards, disability is determined when the disability becomes fixed after a certain period of treatment, except for brain lesion disorders. Every three years, the Ministry of Health and Welfare and the Korea Institute for Health and Social Affairs conduct a survey to gather accurate data on the living conditions and utilization of welfare services by individuals with disabilities in the country [2]. These data are used as basic data for establishing and implementing Korea’s short- and long-term disability welfare policies [2]. According to the 2020 Survey on People with Disabilities, approximately 2.6 million individuals with disabilities are officially registered in Korea. Among them, approximately 58,000 are adolescents aged 12 to 18 with physical or mental impairments or disabilities [2]. Adolescence is a crucial phase for identity formation, and for adolescents with disabilities, it can bring about a complex array of issues such as physical limitations, psychological conflicts, and social isolation [3]. Consequently, it is imperative to provide them with heightened care and attention during this pivotal period.

Physical leisure activities refer to cultural and recreational activities that involve physical movement. It is known that the physical leisure activities of adolescents with a disability are low due to personal, social, and policy deficiencies. However, research results show that the disabled youth’s preference for physical leisure activities is no different from that of the non-disabled youth [4]. Physical leisure activities play a vital role in the holistic development and well-being of adolescents, including those with disabilities [5]. Engaging in such activities contributes to physical health, social interaction, emotional well-being, and overall life satisfaction. It also has a positive impact on the rehabilitation of students with disabilities [5]. Encouraging activities of daily living further enhances the independence and quality of life for adolescents [4,6]. Many adolescents with physical disabilities have real difficulties and limitations in actively participating in physical activities in the school environment, as they have difficulty performing even fundamental activities of daily living, such as moving or eating [7]. It is therefore important to recognize the individual needs and capabilities of each adolescent and tailor activities accordingly.

Life satisfaction refers to how individuals subjectively evaluate their overall quality of life [4]. It encompasses the experience of positive emotions and joy acquired from daily activities, which help to create positive self-awareness and values [6]. Particularly during adolescence, life satisfaction is a significant measure of mental and physical well-being, as well as resilience [8]. Notably, among adolescents, life satisfaction is linked to the quality of social relationships, a positive self-concept, and even academic performance [9]. Fundamentally, it represents the psychological factors that influence an individual’s well-being and their ability to fulfil social and economic duties in their daily life [10,11]. The capacity to lead an independent life as a human being is critical in determining one’s level of life satisfaction [11].

Activities of daily living (ADLs) are a concept that includes personal hygiene such as washing one’s face, brushing teeth, and bathing, activities related to elimination such as using the toilet and controlling bowel movements, and light movements such as getting up, sitting down, and getting out of the room [12]. It is a fundamental and crucial aspect of daily life satisfaction [13]. Enhancing the capacity of disabled individuals to carry out daily activities not only promotes their stability and independence but also fosters self-reliance and self-esteem, enabling them to engage in activities according to their preferences [13]. Previous research has consistently demonstrated that ADLs significantly impact overall life satisfaction [14,15]. Specifically, young individuals with disability encounter challenges related to daily living functions, such as mobility [16]. Consequently, disability-induced limitations in daily life diminish their life satisfaction [17]. The decline in life satisfaction not only undermines individuals with disabilities’ will to live but also diminishes their overall life satisfaction [11]. However, since disability implies long-term impairments rather than temporary limitations, improving the life satisfaction of children and adolescents with physical disabilities cannot be achieved only by improvements in their daily life activities. Consequently, rather than focusing exclusively on improving ADLs, efforts are being undertaken to improve these individuals’ life satisfaction by increasing their engagement in physical leisure activities [18,19].

However, participating in leisure activities is not easy for adolescents [20,21]. Individuals with disabilities are significantly less likely to participate in physical leisure activities [22]. For adolescents with physical disabilities, studies indicate that engagement in recreational physical activities can be enhanced despite limitations in daily functioning [23]. However, research on the significance of leisure activities in the relationship between ADLs and life satisfaction in adolescents with disabilities is limited. There are some potential reasons why this area of research might be limited are the diverse nature of disabilities, focus on the medical aspect, and methodological challenges. Among other reasons, adolescents with disabilities are a more vulnerable group, making it difficult to conduct research; even if data are obtained, research has limits, such as ensuring research reliability.

Understanding the potential interplay between ADLs, physical leisure activities, and life satisfaction is essential for developing comprehensive interventions and support systems for adolescents with disabilities. Since 1980, the Korean Ministry of Health and Welfare and the Korean Institute for Health and Social Affairs have collected cross-sectional data from a sample of individuals with disabilities of all ages, every three years [2]. This study investigated the influence of ADLs and physical leisure activities on the life satisfaction of adolescents with disabilities using these data. Furthermore, the role of leisure activities in the relationship between life satisfaction and ADLs was explored.

## 2. Materials and Methods

### 2.1. Study Design

A cross-sectional study design with secondary data analysis was adopted. Data from the 2020 National Survey of Disabled Persons (NSDP) conducted in 2020 for adolescents with disabilities were utilized.

### 2.2. Data Source and Participants

The NSDP data involved face-to-face interviews with 7025 disabled individuals across Korea, conducted between October 2020 and February 2021. To ensure the reliability of the collected data, weighting was applied to reflect the geographical distribution and types of disabilities among the registered individuals. Additionally, a post-stratification method was employed to address estimation bias.

In this study, an analysis was performed on a subset of the weighted data obtained from the 2020 NSDP. The inclusion criteria were students aged 12 to 18 years, physical disability, and attendance at middle school or high school. Physical disabilities, brain lesions, visual impairments, hearing impairments, speech impairments, and facial impairments were all recognized as external physical dysfunctions. Exclusion criteria were (1) age under 12 or over 19 years, (2) intellectual disability or internal disabilities, and (3) attending elementary school or not attending school. Weights were assigned to the data to represent geographic location and specific types of disability. The total number of weighted study participants in this study was 5364 adolescents (Figure 1).

### 2.3. Study Variables

Demographics included sex, education level (middle, high school), school type (general school–general class, general school–special class for the disabled, school for the disabled), receipt of medical benefit (yes, no), continuous medical treatment (yes, no), having a chronic disease (yes, no), perceived health status (bad, moderate, good), internet usability (yes, no), feeling sad or hopeless, severity of disability (severe, mild disability), ability to go out alone (yes, no), and whether help was needed from others (yes, no).

The ability to execute diverse activities independently determines ADLs. Changing clothes, bathing, oral hygiene, feeding, eating, changing body position while lying down, mobility, sitting, maintaining a seated position, walking, moving about, defecation, and urination are tasks covered in ADLs. To calculate the overall score for these 12 items, a value is assigned to each question based on the level of support required: “no support” is 0 points, “some support required” is 1 point, “considerable support required” is 2 points, and “full support required” is 3 points. To calculate the total scores, all items were reverse-transformed. The total score ranged from 0 to 36, with higher scores indicating greater autonomy in ADLs.

Life satisfaction included nine items for the question, “How satisfied are you with your current life?” Life satisfaction among people with disabilities was measured using a set of nine items that included factors such as familial relationships, social connections, living situations, health status, monthly income, leisure activities, occupation, marital status, and overall current life situation. Six of the nine categories were examined for adolescents after eliminating items that were not relevant to their specific context, such as monthly income, employment progress, and marital status. Each item was scored on a 4-point Likert scale ranging from “very satisfied” (1 point) to “very unhappy” (4 points). All items were reverse-transformed to calculate the total scores. The overall score was calculated by summing the scores for each item, yielding a score range of 6–24 points. Higher scores indicated greater life satisfaction.

Physical leisure encompasses various activities involving physical movement within cultural and recreational pursuits. A survey was conducted to gather information about individuals with disabilities, focused on determining their engagement in activities over the past week. The survey encompassed different areas such as cultural and artistic experiences (attending plays, movies, music concerts, art exhibitions, etc.), gaming (playing games like go, billiards, horse racing, etc.), active participation in cultural and artistic endeavors (participating in reading discussions, writing, calligraphy, musical instrument performances, photography, etc.), personal hobbies and self-development activities (cooking, reading, pursuing technical qualifications, learning English, taking liberal arts classes, etc.), sports (soccer, tennis, swimming, etc.), social (volunteer) service, religious activities, travel (tourism, mountain climbing, fishing, hiking, etc.), as well as other activities such as overseas travel within the past year, socializing with friends and relatives, engaging in family-related outings (eating out, shopping, visiting weekend farms, etc.), and taking time to rest (e.g., sauna visits). A score of 0 was assigned if the individual did not engage in any of these activities, whereas a score of 1 was assigned to a “yes” response, indicating direct involvement. The overall score ranged from 0 to 11, with higher scores indicating a higher level of engagement in physical leisure activities.

### 2.4. Statistical Analysis

The following analyses were performed using the SPSS WIN 24.0 program for this study. Descriptive statistics were employed to analyze the general characteristics and study variables. Pearson’s correlation coefficient analysis was used to examine correlations between ADLs, life satisfaction, and physical leisure activities. Finally, Baron and Kenny’s method (multivariate regression) [23] was used to identify the moderating effect of physical leisure activities on the relationship between ADLs and life satisfaction.

Baron and Kenny’s approach [23] involves a three-step hierarchical regression analysis to examine the moderating effect. In the regression analysis conducted for the current study’s objectives, life satisfaction served as the dependent variable. In the first step, ADL was entered as the independent variable. The second step included the addition of the moderating variable, physical leisure activity, and the third step involved inputting the interaction variable, obtained by multiplying the independent variable and the moderating variable. Mean centering (the value of each data point minus the mean value of that variable) was employed to eliminate multicollinearity with the independent and moderating variables.

## 3. Results

### 3.1. General Characteristics

Table 1 presents the general characteristics of 5365 adolescents with disabilities. Among them, 45.9% were female and 47.5% were middle school students. The majority (78.2%) of participants attended regular classes in general schools. Chronic diseases were absent in 71.4% of participants, and 87.8% had internet access. Only 3.4% of participants experienced feelings of sadness or hopelessness. In terms of disability, 41.3% had mild disabilities and 77.8% could go out independently. On average, they scored 41.79 ± 11.51 (range 14.0–48.0) points in ADLs, 12.56 ± 2.24 (range 7.0–19.0) points in life satisfaction, and 1.27 ± 2.00 (0.0–8.0) points in physical leisure activities.

### 3.2. Comparison of Physical and Psychological Factors

ADLs were significantly positively correlated with physical leisure activities (r = 0.21, *p* < 0.001) as well as life satisfaction (r = 0.35, *p* < 0.001). Physical leisure activities had a significant positive correlation with life satisfaction (r = 0.35, *p* < 0.001). There was a significant positive correlation between physical leisure activities and life satisfaction (r = 0.22, *p* < 0.001) (Table 2).

### 3.3. Moderating Effect of Physical Leisure Activities Relationship between ADLs and Life Satisfaction

Table 3 presents the findings regarding the moderating role of physical leisure activities in the association between ADL and life satisfaction. A moderating three-step hierarchical regression analysis using Baron and Kenny’s method was conducted to investigate this moderating effect. The impact of the independent variable, ADLs, on life satisfaction was first assessed. Physical leisure activity, which was a moderating variable, was introduced as an additional independent variable in the second stage to determine its influence on life satisfaction. Finally, the independent variables, ADLs and physical leisure activity, were multiplied in the third stage to explore their combined effects on life satisfaction.

ADLs accounted for 12.1% of life satisfaction in the first stage of the hierarchical regression model, and the relationship was statistically significant (F = 741.37, *p* < 0.001). Life satisfaction was significantly associated with ADLs (β = 0.35, *p* < 0.001). Physical leisure activities increased the explanatory power to 14.4% (F = 452.01, *p* < 0.001) in the second stage. Physical leisure activities (β = 0.15, *p* < 0.001) and ADLs (β = 0.32, *p* < 0.001) were identified as influential variables for life satisfaction. A moderating variable that compounded the effects of ADLs and physical leisure activities was added in the third stage. When the interaction between ADLs and physical leisure activities was included, the coefficient of determination increased significantly by 6.5% (*p* < 0.001), resulting in an overall explanatory power of 20.9%. Life satisfaction was influenced by the interaction (β = 0.65, *p* < 0.001) between ADLs (β = 0.85, *p* < 0.001) and physical leisure activities (β = −0.24, *p* < 0.001). Physical leisure activity was found to moderate the effect of ADLs on life satisfaction, as the coefficient of determination increased significantly, and the interaction (*p* < 0.001) was significant. 

Examining Figure 2, designed to validate the interaction pattern, it becomes evident that a heightened engagement in physical leisure activities is associated with a more pronounced correlation curve between ADLs and life satisfaction. In other words, the positive correlation between ADLs and life satisfaction strengthens with an increase in physical leisure activities.

## 4. Discussion

This study discovered that although ADLs had a positive effect on life satisfaction, physical leisure activities had a positive effect on life satisfaction among adolescents with disabilities. Physical leisure activities moderated the relationship between ADLs and life satisfaction, thereby indicating the role of physical leisure activities.

This suggests that physical leisure activities moderate life satisfaction, which is influenced by ADLs, and eventually improves life satisfaction. According to prior research, even though adolescents rely heavily on others for daily tasks, their life satisfaction could be enhanced through physical leisure activities improving their life satisfaction [14,15]. Adolescents benefit from physical leisure activities in terms of both mental and physical health at young ages [20]. Engaging in physical leisure activities can have a positive impact on life satisfaction. Regular physical activity has been associated with various physical and mental health benefits, including improved mood, reduced stress, enhanced cognitive function, and better overall well-being in general [19]. For adolescents with disabilities, physical leisure activity can also provide additional benefits such as therapeutic effects, improved psychological well-being, and increased social participation [5,22,24,25]. Despite the modest number of students with low levels of sadness and hopelessness, the life satisfaction score of adolescents with disabilities in this study was found to be lower than that in a previous study, at roughly 12 points (ranging from 7 to 19) [3]. Through this, we can confirm that the life satisfaction of youth with physical limitations is lower than that of youth with disabilities overall. It is generally known that adolescents with disabilities have reduced life satisfaction for various reasons, such as limitations in daily life and personal leisure activities, but it is believed that more follow-up research should be conducted on the reasons.

This study’s data showed that low life satisfaction among disabled adolescents was associated with low physical leisure activities. Adolescents with disabilities have a decline in ADLs because of physical–cognitive impairment depending on the type of disability [17]. Among the factors affecting life satisfaction, low ADL causes low life satisfaction [5]. ADLs are essential self-care tasks that individuals need to perform daily for their well-being [6]. These activities can include tasks such as dressing, bathing, eating, and mobility [6]. The ability to independently carry out ADLs is closely linked to an individual’s overall functional capacity and can influence their life satisfaction [17]. When people participate in physical leisure activities, they often experience improvements in their physical fitness and functional abilities. These improvements can, in turn, positively influence their performance in ADLs. For example, someone who engages in regular activities may experience increased strength, flexibility, and endurance, making it easier for them to perform daily tasks [18].

On the other side, studies have reported that adolescents with disabilities have difficulties in physical leisure activities due to physical, social, cognitive, and emotional limitations that limit their daily lives [5,22,26,27]. In other words, the degree of ADL and the level of leisure activities can affect life satisfaction in adolescents with disabilities. However, considering that the ADLs of adolescents with disabilities is not a factor that can be easily changed, it is critical to investigate the role of physical leisure activities in enhancing the life satisfaction of those adolescents with disabilities in school. This study showed that the degree of physical leisure activity scores in adolescents with disabilities was relatively low (1.27 out of 11 points) compared to studies examining leisure activities among the Korean youth [24]. Similar to non-disabled adolescents, computer games (75%) were played the most, and 23% of students were participating in sports, with less than 5% for traveling, art viewing, or participation. These tasks necessitate the assistance of another person or infrastructure in terms of social awareness and political support, beyond their personal willingness [5,21,22,28].

According to the inclusive school education policy implemented since 2013 in Korea, disabled youths are currently receiving education from a general school, and the percentage has reached 78% in these data. Although students with severe disabilities were included, their ADL level was above average, so they were able to take classes with non-disabled students. This means that participation in general physical activities is possible only if the ADL level is moderate or higher, which means that participation in physical leisure activities may be limited for Korean disabled youth with low ADL levels. Several issues and implications should be considered to improve the engagement of adolescents with disabilities in physical leisure activities. First, there is a need to cultivate enthusiasm and interest among physically impaired adolescents to participate in such activities. Previous studies [5] have shown that children’s preferences, friendships, and enjoyment are factors that promote participation in leisure activities. In addition, it is essential to consider the influence of the immediate social environment, including parents, peers, and program operators, on fostering participation [5]. Second, ensuring accessibility to program operation venues and maintaining suitable facility environments are integral parts of a comprehensive strategy aimed at increasing physical leisure activities among adolescents with disabilities [25]. As adolescents with disabilities become more capable in their ADLs and experience the physical and mental benefits of leisure activities, it can contribute to an overall sense of well-being and life satisfaction. It is important to note that individual preferences and health conditions vary, so the specific activities that contribute to life satisfaction may differ from person to person. Additionally, factors such as social interactions, environmental factors, and personal goals can also play a role in the relationship between physical leisure activities, ADLs, and life satisfaction. This positive cycle, where physical leisure activities contribute to improved ADLs and, consequently, enhanced life satisfaction, highlights the interconnectedness of physical activity and overall quality of life.

The interpretation of the national data used in this study has certain limitations. Although representative sample analysis results from cross-sectional surveys were employed, generalization and interpretation of causal relationships between variables over time are limited. Another limitation was the scarcity of ADL information about school life. Since the ADLs skills necessary for school life cannot be measured, information may be limited because collected ADL scores are utilized. The survey was not conducted since there were no questions about individuals’ willingness to engage in physical leisure activities.

Nonetheless, engaging in physical leisure activities serves as a buffer between ADLs and life satisfaction. Another advantage of this study is that the results can be generalized to some extent for adolescents with physical disabilities utilizing weighted data from a large sample. Tailoring physical leisure activities based on the ADL level and disability type of disabled youth is a nuanced and personalized approach that can have significant implications for their overall well-being. There are some considerations for intervention studies. First, designing interventions that consider the specific needs, challenges, and capabilities of disabled youth based on their ADL level and disability type allows for a more personalized approach. Second, understanding the unique requirements of disabled youth can help to create interventions that are more inclusive and accommodating. Third, investigating the relationship between physical leisure activities and specific ADLs can provide insights into how these activities directly influence daily functioning. As technology continues to advance, exploring the integration of assistive technologies and virtual platforms in tailored interventions can open up new possibilities for engagement and accessibility.

## 5. Conclusions

It was concluded that physical leisure activities play a moderating role in the relationship between activities of daily living and life satisfaction among adolescents with disabilities. The results demonstrate a significant correlation between physical leisure activities, everyday living activities, and life satisfaction. Consequently, these findings emphasize the importance of developing a program that promotes physical leisure activities that are specifically designed for adolescents with physical impairments. The findings of this study can be used to develop future programs focused at improving physical leisure activities for adolescents with a physical impairment. Furthermore, it presents a program that allows adolescents with limitations to participate in physical recreational activities while minimizing environmental constraints. This can be accomplished by utilizing modern technology like AI robots, augmented reality (AR), and virtual reality (VR), which are currently in development. In conclusion, future research that explores the intersection of physical leisure activities, ADLs, and disability types among adolescents with disabilities has the potential to provide targeted and evidence-based interventions. This approach may not only improve the overall quality of life for disabled youth but also contribute valuable insights to the broader field of disability research and healthcare.

## Figures and Tables

**Figure 1 healthcare-12-00190-f001:**
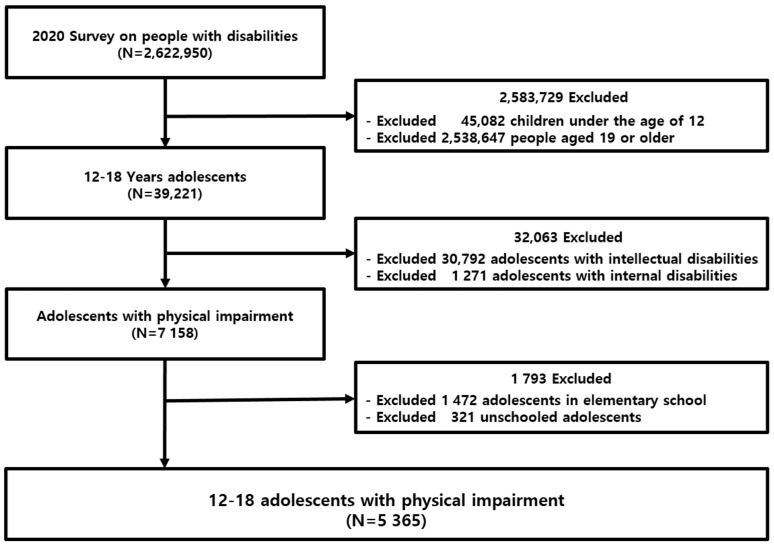
Flowchart diagram of data extraction.

**Figure 2 healthcare-12-00190-f002:**
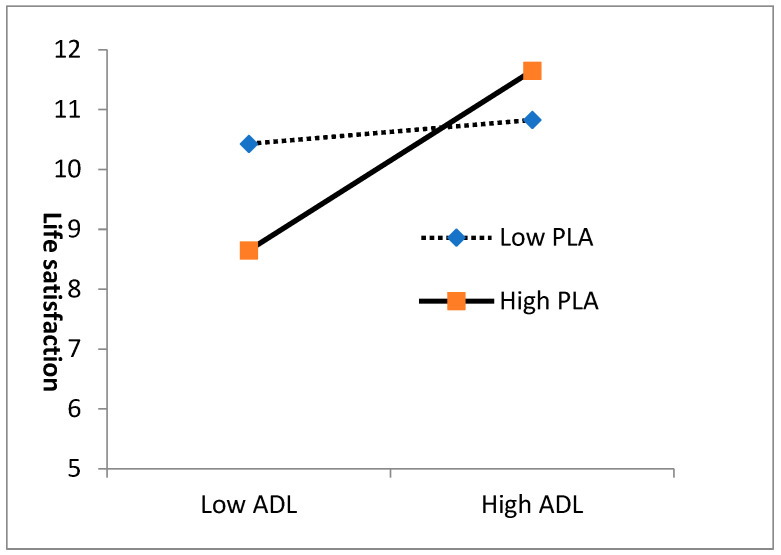
The two-way interactions: the slope of the relationship between the ADLs and life satisfaction is changed by the physical leisure activities (PLAs).

**Table 1 healthcare-12-00190-t001:** General characteristics (N = 5365).

Variables	Categories	*n* (%)
Sex	Female	2464 (45.9)
	Male	2900 (54.1)
Education level	Middle School	2547 (47.5)
	High school	2818 (52.5)
School type	General school	4195 (78.2)
	Disabled class in general school	532 (9.9)
	Disabled school	637 (11.9)
Receipt of medical benefit	Yes	79 (1.5)
	No	5285 (98.5)
Continuous treatment	Yes	4487 (83.6)
	No	877 (16.4)
Having chronic diseases	Yes	1534 (28.6)
	No	3831 (71.4)
Perceived health status	Bad	1373 (25.6)
	Moderate	1982 (36.9)
	Good	2010 (37.5)
Internet use availability	Yes	4709 (87.8)
	No	656 (12.2)
Feeling sad or hopeless	Yes	184 (3.4)
	No	5181 (96.6)
Degree of disability	Severe disability	2214 (41.3)
	Mild disability	3151 (58.7)
Ability to go out alone	Yes	4176 (77.8)
	No	1189 (22.2)
Whether need help from others	Yes	1283 (23.9)
	No	4082 (76.7)

**Table 2 healthcare-12-00190-t002:** Correlation among activities of daily living, life satisfaction, and physical leisure activities (N = 5365).

	Activities of Daily Living r (*p*)	Life Satisfaction r (*p*)
Life satisfaction	0.35 (<0.001)	1
Physical leisure activities	0.21 (<0.001)	0.22 (<0.001)

**Table 3 healthcare-12-00190-t003:** Moderating effect of physical leisure activities’ relationship between ADL and life satisfaction (N = 5365).

Step	Variables	B	SE	*β*	t	*p*	Adjusted. R²	F (*p*)
1	Constant	14.59	0.11		134.90	<0.001	0.121	741.37 (<0.001)
	ADL	0.07	0.00	0.35	27.23	<0.001		
2	Constant	14.63	0.11		136.99	<0.001	0.144	452.01 (<0.001)
	ADL	0.06	0.00	0.32	24.49	<0.001		
	PLA	0.17	0.01	0.15	11.96	<0.001		
3	Constant	10.39	0.23		45.93	<0.001	0.209	473.96 (<0.001)
	ADL	0.17	0.00	0.85	30.13	<0.001		
	PLA	−0.27	0.03	−0.24	−10.66	<0.001		
	ADL×PLA	0.10	0.01	0.65	21.06	<0.001		

ADL: activities of daily living; PLA: physical leisure activities. The difference value between step 2 and step 3 was △R^2^ = 0.065 (*p* < 0.001).

## Data Availability

The data presented in this study are available upon request from the corresponding author. The data are not publicly available because of privacy concerns.

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
