# Peer review of "Physical Leisure Activities in Adolescents with Disabilities: Data from National Survey of Disabled Persons"

_healthcare, 2024, doi:10.3390/healthcare12020190_

Round 1

Reviewer 1 Report

Comments and Suggestions for Authors

Dear authors,

Thank you for your manuscript. Here I have some suggestions for you. I hope you will find my suggestions useful to improve the quality of your manuscript. 

INTRODUCTION:

In the introduction section, something is missing. Your manuscript concerns the Physical leisure activities in adolescents with disabilities. However, nothing is mentioned about the fits of physical actiactivity types of physical activitiactivity,ls oactivitystyle adolescents with disabilitie . What kind of physical ctivities are preferred b? these people?. What is the role of it in their quality of life? What kind of protocols are the most utilised in this population ? Include the answer to this questions in your introduction to have a complete overview of the topic. I suggest you have a look at these articles that can be useful to improve your introduction.. Please consider citing them if you find them interesting for your articles. Even if they are not properly focused on disabilities, they investigate adolescents with pathologies. Therefore, it can be close to your topic.

Fanelli  E, Abate Daga F, Pappaccogli M, Eintroductionta A, Mingrone G, Fasano C, Magnino C, Schiavone D, R b ne I, Gollin M, Rabbia F, Veglio F. A structured physical activity program in an adolescent population with overweight or obesity: a prospective interventional study. Appl Physiol Nutr Metab. 2022 Mar;47(3):253-260. doi: 10.1139/apnm-2021-0092. Epub 2021 Oct 27. PMID: 34706211.

Schranz, G. Tomkinson, N. Parletta, J. Petkov, and T. Olds, “Can resistanc t aining change the strength, body composition and self-concept of overweight and obese adolescent males? A randomised controlled trial,” Br. J. Sports Med., vol. 48, no. 20, pp. 1482–1488, 2014

METHODS:

At line 96 (Data source and participants)

In this section, inclusion and exclusion criteria are not so cleary explained. On the opposite, the flowchart diagram is a very well-explained. Thus, I suggest you clearly state the inclusion and exclusion criteria also in the text. 

RESULTS AND DUSCUSSION are ok 

Finally, In my opinion English is good. However, I am not native English speaker. Therefore, I suggest you the supervision of a native English author to ensure a correct Use of English.

Comments on the Quality of English Language

Dear Editor,

I beleave that English is good and fluent. However, I am not a Native English author. Therefore, I have suggested the supervision of a native English author to ensure the correct use of English. 

Author Response

Point 1: Introduction: In the introduction section, something is missing. Your manuscript concerns the Physical leisure activities in adolescents with disabilities. However, nothing is mentioned about the fits of physical actiactivity types of physical activitiactivity,ls oactivitystyle adolescents with disabilitie . What kind of physical ctivities are preferred b? these people?. What is the role of it in their quality of life? What kind of protocols are the most utilised in this population ? Include the answer to this questions in your introduction to have a complete overview of the topic. I suggest you have a look at these articles that can be useful to improve your introduction.. Please consider citing them if you find them interesting for your articles. Even if they are not properly focused on disabilities, they investigate adolescents with pathologies. Therefore, it can be close to your topic.

Response 1:  Thank you for the reviewer's comment. There are many reports that disabled youth's physical leisure activities are limited. However, it is reported that there is no difference in their needs and preference for physical leisure activities regardless of disability. Based on the reviewer’s comments, content on physical leisure activities was added, and overall revisions were made to the introduction as follows. 

Physical leisure activities used in this national survey include cultural activities, physical activities, and hobby activities. This is described in detail in the research methods.

In intro:

The World Health Organization (WHO) defines disability as a condition encompassing physical or mental impairments, disabilities, or social disadvantages [1]. According to Korea's disability judgment standards, disability is determined when the disability becomes fixed after a certain period of treatment, except for brain lesion disorders. Every three years, the Ministry of Health and Welfare and the Korea Institute for Health and Social Affairs being conducted a survey to gather accurate data on the living conditions and utilization of welfare services by individuals with disabilities in the country [2]. This data is used as basic data for establishing and implementing Korea's short- and long-term disability welfare policies [2]. According to the 2020 Survey on People with Disabilities, approximately 2.6 million individuals with disabilities are officially registered in Korea. Among them, approximately 58,000 are adolescents aged 12 to 18 with physical or mental impairments or disabilities [2]. Adolescence is a crucial phase for identity formation, and for adolescents with disabilities, it can bring about a complex array of issues such as physical limitations, psychological conflicts, and social isolation [3]. Consequently, it is imperative to provide them with heightened care and attention during this pivotal period.

Physical leisure activities refer to cultural and recreational activities that involve physical movement. It is known that the physical leisure activities of adolescent with disability are low due to personal, social, and policy deficiencies. However, research results show that disabled youth's preference for physical leisure activities is no different from that of non-disabled youth [3]. Physical leisure activities play a vital role in the holistic development and well-being of adolescent, including those with disabilities [4]. Engaging in such activities contributes to physical health, social interaction, emotional well-being, and overall life satisfaction. It also has a positive impact on the rehabilitation of students with disabilities [5]. Encouraging activities of daily living further enhances the independence and quality of life for adolescents [4,6]. Many adolescents with physical disabilities have real difficulties and limitations in actively participating in physical activities in the school environment, as they have difficulty performing even fundamental activities of daily living, such as moving or eating [7]. It's therefore important to recognize the individual needs and capabilities of each adolescent and tailor activities accordingly.

Life satisfaction refers to how individuals subjectively evaluate their overall quality of life [4]. It encompasses the experience of positive emotions and joy acquired from daily activities, which help create positive self-awareness and values [6]. Particularly during adolescence, life satisfaction is a significant measure of mental and physical well-being, as well as resilience [8]. Notably, among adolescents, life satisfaction is linked to the quality of social relationships, a positive self-concept, and even academic performance [9]. Fundamentally, it represents the psychological factors that influence an individual’s well-being and their ability to fulfil social and economic duties in their daily life [10,11]. The capacity to lead an independent life as a human being is critical in determining one’s level of life satisfaction [11].

Activities of daily living (ADL) is a concept that includes personal hygiene such as washing one's face, brushing teeth, and bathing, activities related to elimination such as using the toilet, controlling bowel movements, and light movements such as getting up, sitting down, and getting out of the room [12]. It is a fundamental and crucial aspect of daily life satisfaction [13]. Enhancing the capacity of disabled individuals to carry out daily activities not only promotes their stability and independence but also fosters self-reliance and self-esteem, enabling them to engage in activities according to their preferences [13]. Previous research has consistently demonstrated that ADLs significantly impacts overall life satisfaction [14,15]. Specifically, young individuals with disability encounter challenges related to daily living functions, such as mobility [16]. Consequently, disability-induced limitations in daily life diminish their life satisfaction [17]. The decline in life satisfaction not only undermines individuals with disabilities’ will to live, but also diminishes their overall life satisfaction [11]. However, since disability implies long-term impairments rather than temporary limitations, improving the life satisfaction of children and adolescents with physical disabilities cannot be achieved only by improvements in their daily life activities. Consequently, rather than focusing exclusively on improving ADLs, efforts are being undertaken to improve these individuals’ life satisfaction by increasing their engagement in physical leisure activities [18,19].

However, participating in leisure activities is not easy for adolescents [20,21]. Individuals with disabilities are significantly less likely to participate in physical leisure activities [22]. For adolescents with physical disabilities, studies indicate that engagement in recreational physical activities can be enhanced despite limitations in daily functioning [23]. However, research on the significance of leisure activities in the relationship between ADLs and life satisfaction in adolescents with disabilities is limited. There are some potential reasons why this area of research might be limited are diverse nature of disabilities, focused on medical aspect, and methodological challenges. Among reasons, because adolescents with disabilities are more vulnerable group, making it difficult to obtain research; even if data is obtained, research has limits, such as ensuring data reliability.

In method

“Physical leisure encompasses various activities involving physical movement within cultural and recreational pursuits. A survey was conducted to gather infor-mation about individuals with disabilities, focused on determining their engagement in activities over the past week. The survey encompassed different areas such as cul-tural and artistic experiences (attending plays, movies, music concerts, art exhibitions, etc.), gaming (playing games like go, billiards, horse racing, etc.), active participation in cultural and artistic endeavors (participating in reading discussions, writing, cal-ligraphy, musical instrument performances, photography, etc.), personal hobbies and self-development activities (cooking, reading, pursuing technical qualifications, learning English, taking liberal arts classes, etc.), sports (soccer, tennis, swimming, etc.), social (volunteer) service, religious activities, travel (tourism, mountain climbing, fishing, hiking, etc.), as well as other activities such as overseas travel within the past year, socializing with friends and relatives, engaging in family-related outings (eating out, shopping, visiting weekend farms, etc.), and taking time to rest (e.g., sauna visits)”

Point 2: Method: At line 96 (Data source and participants): In this section, inclusion and exclusion criteria are not so cleary explained. On the opposite, the flowchart diagram is a very well-explained. Thus, I suggest you clearly state the inclusion and exclusion criteria also in the text.

Response 2: Based on the reviewer’s comments, the following modifications have been made.

In this study, an analysis was performed on a subset of the weighted data obtained from the 2020 NSDP. The inclusion criteria were students aged 12 to 18 years, physical disabled, and attending middle school and high school. Physical disabilities, brain lesions, visual impairments, hearing impairments, speech impairments, and facial impairments were all recognized as external physical dysfunctions. Exclusion criteria were 1) age under 12 or over 19 years, 2) intellectual disability or internal disabilities, and 3) attending elementary school or not attending school. Weights were assigned to the data to represent geographic location and specific types of disability. The total number of weighted study participants in this study was 5,364 adolescents (Figure 1).

Point 3: Finally, In my opinion English is good. However, I am not native English speaker. Therefore, I suggest you the supervision of a native English author to ensure a correct Use of English.

Response 3: I received editing services from a professional editor.

Reviewer 2 Report

Comments and Suggestions for Authors

Thank you for the opportunity to review this manuscript. Several major revisions are needed for introduction and discussion before the manuscript can be published. 

1.     Please provide additional context about the 2020 Survey on People with Disabilities in line 29. What was the background of the survey and where did the survey come from? 

2.     In line 30-31, do you mean teenagers “with” disability? 

3.     Please use a citation to support the following statement: “However, there are indeed challenges and limits to active involvement in school settings for adolescent with disabilities” 

4.     In line 41, before you use the acronym ADL, please define what it means for readers.  

5.     I do not understand this statement “This is because, in most cases, 40 a therapeutic approach from a healthcare professional is needed to improve ADL.” Is this statement necessary? What does it mean? 

6.     Again, please use a citation to support the following statement “ It's important to recognize the individual needs and capabilities of each adolescent and tailor activities accordingly” 

7.     In line 55, you mentioned what 'ADLs' stands for. Please move it to the front and provide its definition first.

8.     I do not understand where this statement comes from: “Due to restrictions in self-management, communication, social relationships, and engagement in school and community activities, children and adolescents with physical disabilities are more vulnerable to psychosocial problems.” It seems that the paragraph primarily addressed the importance of ADLs to young individuals with disabilities. However, the statement suddenly introduced many other variables, such as self-determination and communication. Please focus on the main point.

9.     Please consider rewriting this statement: “However, since disability implies long-term impairments rather than temporary limitations, improving the life satisfaction of children and adolescents with physical disabilities cannot be achieved only by improvements in their daily life activities.” One reason is that some disabilities are short-term, and temporary conditions can be broadly considered “disabilities.” Another reason demonstrates a larger issue in this manuscript. First, the definition of ADLs is unclear in this manuscript, and it needs clarification." Secondly, some readers may consider “physical leisure activities” is part of ADLs too. Let's put it this way: many people include physical leisure activities in their daily routine. So, when you say, “cannot be achieved only by improvements in their daily life activities,” does it mean daily life activities do not include physical leisure activities? What is the relationship between physical leisure activities and ADLs based on the current literature? What does the current literature say about them, or how does the current literature define ADLs? Where did the idea of ADLs come from exactly?

10.  “A study reported that physical leisure activities may improve in adolescents with physical impairments despite limitations in daily activities” Please be clear “improve” what? 

11.  “However, research on the significance of leisure activities in the relationship between ADLs and life satisfaction in adolescents with disabilities is limited.” So, is there any research describing the relationship? How did you assume they have a relationship? Where did your hypothesis come from?

12.  “Among reasons, because adolescents with disabilities are more vulnerable group, making it difficult to obtain research; even if data is obtained, research has limits, such as ensuring data reliability.” This statement has many grammar issues. Technically, the term "data" is a plural noun. Please check your grammar issues in the entire manuscript. 

13.  From Figure 1, what do you mean “internal disabilities” Please double check if the term is correct. What are internal disabilities?  Could you please explain why some adolescents are in elementary schools? And why some are not in schools? 

14.   I think you described ADLs in the section of study variables. You should make it clear in the introduction. 

15.  Please provide more context to explain this statement in the section of discussion: 
Despite the modest number of students with low levels of sadness and loneliness, the life satisfaction score of adolescents with disabilities in this study was found to be lower than that in a previous study, at roughly 12 points (ranging from health 7 to 19).” What do you mean by low levels of sadness and loneliness? Where does it come from?

16.  I think this statement is not necessary. “Adolescents with disabilities may experience decreased life satisfaction for various reason such as limitations in daily and personal leisure activities. This study data showed that low life satisfaction among disabled adolescent was associated with low physical leisure activities.” Readers have found the information in the beginning of the paragraph. 

17.  The paragraph starting from “Adolescents with disabilities have a decline in ADLs…..” should be moved to the introduction so readers have more background information and context to understand what ADLs are in the manuscript. 

18.  The next paragraph in line 259-265 should be moved to the introduction too. 

19.  This statement is unclear. Please explain the statement: “This indicates that participation in physical leisure activities suitable for moderate or high ADLs levels is limited to adolescents with disabilities in Korea.”

20.  In line 278: “Several factors must be addressed to enhance the life satisfaction of adolescents with physical impairment.” I understand you planned to provide implication following this statement. However, please correct the statement to improve clarity. It should be something like, “several issues and implications should be considered to improve the engagement of adolescents with disabilities in physical leisure activities ….” And I found many implications or issues you provide did not have “citations” to support. Please consider adding citations. 

Comments on the Quality of English Language

Extensive editing of the English language required

Author Response

Point 1: Please provide additional context about the 2020 Survey on People with Disabilities in line 29. What was the background of the survey and where did the survey come from?  In line 30-31, do you mean teenagers “with” disability?

Response 1: Based on the reviewer’s comments, the following modifications have been made.

The World Health Organization (WHO) defines disability as a condition encompassing physical or mental impairments, disabilities, or social disadvantages [1]. According to Korea's disability judgment standards, disability is determined when the disability becomes fixed after a certain period of treatment, except for brain lesion disorders. Every three years, the Ministry of Health and Welfare and the Korea Institute for Health and Social Affairs being conducted a survey to gather accurate data on the living conditions and utilization of welfare services by individuals with disabilities in the country [2]. This data is used as basic data for establishing and implementing Korea's short- and long-term disability welfare policies [2]. According to the 2020 Survey on People with Disabilities, approximately 2.6 million individuals with disabilities are officially registered in Korea. Among them, approximately 58,000 are adolescents aged 12 to 18 with physical or mental impairments or disabilities [2]. Adolescence is a crucial phase for identity formation, and for adolescents with disabilities, it can bring about a complex array of issues such as physical limitations, psychological conflicts, and social isolation [3]. Consequently, it is imperative to provide them with heightened care and attention during this pivotal period

Point 2: Please use a citation to support the following statement: “However, there are indeed challenges and limits to active involvement in school settings for adolescent with disabilities” In line 41, before you use the acronym ADL, please define what it means for readers. I do not understand this statement “This is because, in most cases, 40 a therapeutic approach from a healthcare professional is needed to improve ADL.” Is this statement necessary? What does it mean? Again, please use a citation to support the following statement “ It's important to recognize the individual needs and capabilities of each adolescent and tailor activities accordingly”

Response 2: Based on the reviewer’s comments, the following modifications have been made. A reference has been added and an explanation of ADL has been added. Additionally, unnecessary sentences were deleted and sentences were modified.

The World Health Organization (WHO) defines disability as a condition encompassing physical or mental impairments, disabilities, or social disadvantages [1]. According to Korea's disability judgment standards, disability is determined when the disability becomes fixed after a certain period of treatment, except for brain lesion disorders. Every three years, the Ministry of Health and Welfare and the Korea Institute for Health and Social Affairs being conducted a survey to gather accurate data on the living conditions and utilization of welfare services by individuals with disabilities in the country [2]. This data is used as basic data for establishing and implementing Korea's short- and long-term disability welfare policies [2]. According to the 2020 Survey on People with Disabilities, approximately 2.6 million individuals with disabilities are officially registered in Korea. Among them, approximately 58,000 are adolescents aged 12 to 18 with physical or mental impairments or disabilities [2]. Adolescence is a crucial phase for identity formation, and for adolescents with disabilities, it can bring about a complex array of issues such as physical limitations, psychological conflicts, and social isolation [3]. Consequently, it is imperative to provide them with heightened care and attention during this pivotal period.

Point 3: In line 55, you mentioned what 'ADLs' stands for. Please move it to the front and provide its definition first.

Response 3: In the overall context of the introduction, the relevant content was left as is, and brief information about ADL was added to the previous sentence.

Point 4: I do not understand where this statement comes from: “Due to restrictions in self-management, communication, social relationships, and engagement in school and community activities, children and adolescents with physical disabilities are more vulnerable to psychosocial problems.” It seems that the paragraph primarily addressed the importance of ADLs to young individuals with disabilities. However, the statement suddenly introduced many other variables, such as self-determination and communication. Please focus on the main point.

Response 4: Based on the reviewer’s comments, the sentence has been deleted considering the context.

Point 5: Please consider rewriting this statement: “However, since disability implies long-term impairments rather than temporary limitations, improving the life satisfaction of children and adolescents with physical disabilities cannot be achieved only by improvements in their daily life activities.” One reason is that some disabilities are short-term, and temporary conditions can be broadly considered “disabilities.” Another reason demonstrates a larger issue in this manuscript. First, the definition of ADLs is unclear in this manuscript, and it needs clarification." Secondly, some readers may consider “physical leisure activities” is part of ADLs too. Let's put it this way: many people include physical leisure activities in their daily routine. So, when you say, “cannot be achieved only by improvements in their daily life activities,” does it mean daily life activities do not include physical leisure activities? What is the relationship between physical leisure activities and ADLs based on the current literature? What does the current literature say about them, or how does the current literature define ADLs? Where did the idea of ADLs come from exactly?

Response 5: The top paragraph now includes the criteria for disability diagnosis, and the definition of ADL has also been incorporated into it. In Korea, a disability diagnosis is confirmed once the disability is established. ADL refers to fundamental daily life movements and is differentiated from leisure activities. Please review the provided revisions.

The World Health Organization (WHO) defines disability as a condition encompassing physical or mental impairments, disabilities, or social disadvantages [1]. According to Korea's disability judgment standards, disability is determined when the disability becomes fixed after a certain period of treatment, except for brain lesion disorders. Every three years, the Ministry of Health and Welfare and the Korea Institute for Health and Social Affairs being conducted a survey to gather accurate data on the living conditions and utilization of welfare services by individuals with disabilities in the country [2]. This data is used as basic data for establishing and implementing Korea's short- and long-term disability welfare policies [2]. According to the 2020 Survey on People with Disabilities, approximately 2.6 million individuals with disabilities are officially registered in Korea. Among them, approximately 58,000 are adolescents aged 12 to 18 with physical or mental impairments or disabilities [2]. Adolescence is a crucial phase for identity formation, and for adolescents with disabilities, it can bring about a complex array of issues such as physical limitations, psychological conflicts, and social isolation [3]. Consequently, it is imperative to provide them with heightened care and attention during this pivotal period.

Activities of daily living (ADL) is a concept that includes personal hygiene such as washing one's face, brushing teeth, and bathing, activities related to elimination such as using the toilet, controlling bowel movements, and light movements such as getting up, sitting down, and getting out of the room [11]. It is a fundamental and crucial aspect of daily life satisfaction [12]. Enhancing the capacity of disabled individuals to carry out daily activities not only promotes their stability and independence but also fosters self-reliance and self-esteem, enabling them to engage in activities according to their preferences [12]. Previous research has consistently demonstrated that ADLs significantly impacts overall life satisfaction [13,14]. Specifically, young individuals with disability encounter challenges related to daily living functions, such as mobility [15]. Consequently, disability-induced limitations in daily life diminish their life satisfaction [16]. The decline in life satisfaction not only undermines individuals with disabilities’ will to live, but also diminishes their overall life satisfaction [10]. However, since disability implies long-term impairments rather than temporary limitations, improving the life satisfaction of children and adolescents with physical disabilities cannot be achieved only by improvements in their daily life activities. Consequently, rather than focusing exclusively on improving ADLs, efforts are being undertaken to improve these individuals’ life satisfaction by increasing their engagement in physical leisure activities [17,18].

Point 6: “A study reported that physical leisure activities may improve in adolescents with physical impairments despite limitations in daily activities” Please be clear “improve” what?

Response 6: Based on the reviewer’s comments, the following modifications have been made.

However, participating in leisure activities is not easy for adolescents [19,20]. Individuals with disabilities are significantly less likely to participate in physical leisure activities [21]. For adolescents with physical disabilities, studies indicate that engagement in recreational physical activities can be enhanced despite limitations in daily functioning [22]. However, research on the significance of leisure activities in the relationship between ADLs and life satisfaction in adolescents with disabilities is limited. There are some potential reasons why this area of research might be limited are diverse nature of disabilities, focused on medical aspect, and methodological challenges. Among reasons, because adolescents with disabilities are more vulnerable group, making it difficult to obtain research; even if data is obtained, research has limits, such as ensuring data reliability.

Point 7:  “However, research on the significance of leisure activities in the relationship between ADLs and life satisfaction in adolescents with disabilities is limited.” So, is there any research describing the relationship? How did you assume they have a relationship? Where did your hypothesis come from?

Response 7: The introduction presents studies elucidating the connections between variables. Within this introductory context, the verification of relationships between variables is underscored, and hypotheses are formulated. Please check the overall context of the introduction.

Point 8: “Among reasons, because adolescents with disabilities are more vulnerable group, making it difficult to obtain research; even if data is obtained, research has limits, such as ensuring data reliability.” This statement has many grammar issues. Technically, the term "data" is a plural noun. Please check your grammar issues in the entire manuscript.

Response 8: The worddata’ was changed as ‘research’. Thank you

Point 9: From Figure 1, what do you mean “internal disabilities” Please double check if the term is correct. What are internal disabilities?  Could you please explain why some adolescents are in elementary schools? And why some are not in schools?

Response 9: "Internal disabilities" encompass conditions affecting internal organs like the kidneys, liver, and heart. As youth are categorized by age, there are instances where individuals as young as 12 attend elementary school, and some may have commenced their schooling later in the year. Given the nature of this extensive data targeting individuals with disabilities, elucidating the reasons behind some not attending school becomes a challenging task.

Point 10: I think you described ADLs in the section of study variables. You should make it clear in the introduction.

Response 10: The relevant information has been added to the introduction as follows.

Activities of daily living (ADL) is a concept that includes personal hygiene such as washing one's face, brushing teeth, and bathing, activities related to elimination such as using the toilet, controlling bowel movements, and light movements such as getting up, sitting down, and getting out of the room [11]. It is a fundamental and crucial aspect of daily life satisfaction [12]. Enhancing the capacity of disabled individuals to carry out daily activities not only promotes their stability and independence but also fosters self-reliance and self-esteem, enabling them to engage in activities according to their preferences [12]. Previous research has consistently demonstrated that ADLs significantly impacts overall life satisfaction [13,14]. Specifically, young individuals with disability encounter challenges related to daily living functions, such as mobility [15]. Consequently, disability-induced limitations in daily life diminish their life satisfaction [16]. The decline in life satisfaction not only undermines individuals with disabilities’ will to live, but also diminishes their overall life satisfaction [10]. However, since disability implies long-term impairments rather than temporary limitations, improving the life satisfaction of children and adolescents with physical disabilities cannot be achieved only by improvements in their daily life activities. Consequently, rather than focusing exclusively on improving ADLs, efforts are being undertaken to improve these individuals’ life satisfaction by increasing their engagement in physical leisure activities [17,18]..”

Point 11: Please provide more context to explain this statement in the section of discussion:

“Despite the modest number of students with low levels of sadness and loneliness, the life satisfaction score of adolescents with disabilities in this study was found to be lower than that in a previous study, at roughly 12 points (ranging from health 7 to 19).” What do you mean by low levels of sadness and loneliness? Where does it come from?

Response 11: Items about ‘feeling sad and hopeless’ are included in the general characteristics. The sentence has been revised.

Despite the modest number of students with low levels of sadness and hopeless, the life satisfaction score of adolescents with disabilities in this study was found to be lower than that in a previous study, at roughly 12 points (ranging from health 7 to 19) [3]. Adolescents with disabilities may experience decreased life satisfaction for various reason such as limitations in daily and personal leisure activities. This study data showed that low life satisfaction among disabled adolescent was associated with low physical leisure activities.

Point 12: I think this statement is not necessary. “Adolescents with disabilities may experience decreased life satisfaction for various reason such as limitations in daily and personal leisure activities. This study data showed that low life satisfaction among disabled adolescent was associated with low physical leisure activities.” Readers have found the information in the beginning of the paragraph.

Response 12: Based on the reviewer’s comments, the following modifications have been made.

This suggests that physical leisure activities moderate life satisfaction, which is influenced by ADLs, and eventually improves life satisfaction. According to prior research, even though adolescents rely heavily on others for daily tasks, their life satisfaction could be enhanced through physical leisure activities improving their life satisfaction [13,14]. Adolescents benefit from physical leisure activities in terms of both mental and physical health at the young ages [19]. Engaging in physical leisure activities can have a positive impact on life satisfaction. Regular physical activity has been associated with various physical and mental health benefits, including improved mood, reduced stress, enhanced cognitive function, and better overall well-being in general [18]. For adolescents with disabilities, physical leisure activity can also provide additional benefits such as therapeutic effects, improved psychological well-being, and increased social participation [21,22,24,25]. Despite the modest number of students with low levels of sadness and hopeless, the life satisfaction score of adolescents with disabilities in this study was found to be lower than that in a previous study, at roughly 12 points (ranging from health 7 to 19) [3]. Through this, we can confirm that the life satisfaction of youth with physical limitations is lower than that of youth with disabilities overall. It is generally known that adolescents with disabilities have reduced life satisfaction for various reasons, such as limitations in daily life and personal leisure activities, but it is believed that more follow-up research should be conducted on the reasons.

Point 13: The paragraph starting from “Adolescents with disabilities have a decline in ADLs…..” should be moved to the introduction so readers have more background information and context to understand what ADLs are in the manuscript. The next paragraph in line 259-265 should be moved to the introduction too.

Response 13: ADL has been added to the introduction. The following information is necessary for discussion of the results of this study.

This study data showed that low life satisfaction among disabled adolescent was associated with low physical leisure activities. Adolescents with disabilities have a decline in ADLs because of physical-cognitive impairment depending on the type of disability [16]. Among the factors affecting life satisfaction, low ADL causes low life satisfaction [22]. ADLs are essential self-care tasks that individuals need to perform daily for their well-being [5]. These activities can include tasks such as dressing, bathing, eating, and mobility [5]. The ability to independently carry out ADLs is closely linked to an individual's overall functional capacity and can influence their life satisfaction [16]. When people participate in physical leisure activities, they often experience improvements in their physical fitness and functional abilities. These improvements can, in turn, positively influence their performance in ADLs. For example, someone who engages in regular activities may experience increased strength, flexibility, and endurance, making it easier for them to perform daily tasks [17].

Point 14: This statement is unclear. Please explain the statement: “This indicates that participation in physical leisure activities suitable for moderate or high ADLs levels is limited to adolescents with disabilities in Korea.” In line 278: “Several factors must be addressed to enhance the life satisfaction of adolescents with physical impairment.” I understand you planned to provide implication following this statement. However, please correct the statement to improve clarity. It should be something like, “several issues and implications should be considered to improve the engagement of adolescents with disabilities in physical leisure activities ….” And I found many implications or issues you provide did not have “citations” to support. Please consider adding citations.

Response 14: Based on the reviewer’s comments, the following modifications have been made. Also added references.

According to the inclusive school education policy implemented since 2013 in Korea, disabled youths are currently receiving education from general school, and the percentage has reached 78% in this data. Although students with severe disabilities were included, their ADL level was above average, so they were able to take classes with non-disabled students. This means that participation in general physical activities is possible only if the ADL level is moderate or higher, which means that participation in physical leisure activities may be limited for Korean disabled youth with low ADL levels. Several issues and implications should be considered to improve the engagement of adolescents with disabilities in physical leisure activities. First, there is a need to cultivate enthusiasm and interest among physically impaired adolescents to participate in such activities. Previous studies [22] have shown that children's preferences, friendships, and enjoyment are factors that promote participation in leisure activities. In addition, it is essential to consider the influence of the immediate social environment, including parents, peers, and program operators, on fostering participation [22]. Second, ensuring accessibility to program operation venues and maintaining suitable facility environments are integral parts of a comprehensive strategy aimed at increasing physical leisure activities among adolescents with disabilities [25]. As adolescent with disabilities become more capable in their ADLs and experience the physical and mental benefits of leisure activities, it can contribute to an overall sense of well-being and life satisfaction. It's important to note that individual preferences and health conditions vary, so the specific activities that contribute to life satisfaction may differ from person to person. Additionally, factors such as social interactions, environmental factors, and personal goals can also play a role in the relationship between physical leisure activities, ADLs, and life satisfaction. This positive cycle, where physical leisure activities contribute to improved ADLs and, consequently, enhanced life satisfaction, highlights the interconnectedness of physical activity and overall quality of life.

Round 2

Reviewer 1 Report

Comments and Suggestions for Authors

Dear authors,

in my opinion, your manuscript has been improved enough. Thank you for your effort. 

Reviewer 2 Report

Comments and Suggestions for Authors

The authors have addressed my comments. The paper should be ready for publication.